# Spectral Clustering of Graphs with the Bethe Hessian

**Alaa Saade**
Laboratoire de Physique Statistique, CNRS UMR 8550
École Normale Superieure, 24 Rue Lhomond Paris 75005

**Florent Krzakala**[*]
Sorbonne Universités, UPMC Univ Paris 06
Laboratoire de Physique Statistique, CNRS UMR 8550
École Normale Superieure, 24 Rue Lhomond
Paris 75005

**Lenka Zdeborová**
Institut de Physique Théorique
CEA Saclay and CNRS URA 2306
91191 Gif-sur-Yvette, France

## Abstract

Spectral clustering is a standard approach to label nodes on a graph by studying the (largest or lowest) eigenvalues of a symmetric real matrix such as e.g. the adjacency or the Laplacian. Recently, it has been argued that using instead a more complicated, non-symmetric and higher dimensional operator, related to the non-backtracking walk on the graph, leads to improved performance in detecting clusters, and even to optimal performance for the stochastic block model. Here, we propose to use instead a simpler object, a symmetric real matrix known as the Bethe Hessian operator, or deformed Laplacian. We show that this approach combines the performances of the non-backtracking operator, thus detecting clusters all the way down to the theoretical limit in the stochastic block model, with the computational, theoretical and memory advantages of real symmetric matrices.

Clustering a graph into groups or functional modules (sometimes called communities) is a central task in many fields ranging from machine learning to biology. A common benchmark for this problem is to consider graphs generated by the stochastic block model (SBM) [7, 22]. In this case, one considers $n$ vertices and each of them has a group label $g_v \in \{1, \ldots, q\}$. A graph is then created as follows: all edges are generated independently according to a $q \times q$ matrix $p$ of probabilities, with $\Pr[A_{u,v} = 1] = p_{g_u, g_v}$. The group labels are hidden, and the task is to infer them from the knowledge of the graph. The stochastic block model generates graphs that are a generalization of the Erdős-Rényi ensemble where an unknown labeling has been hidden.

We concentrate on the sparse case, where algorithmic challenges appear. In this case $p_{ab}$ is $O(1/n)$, and we denote $p_{ab} = c_{ab}/n$. For simplicity we concentrate on the most commonly-studied case where groups are equally sized, $c_{ab} = c_{\text{in}}$ if $a = b$ and $c_{ab} = c_{\text{out}}$ if $a \neq b$. Fixing $c_{\text{in}} > c_{\text{out}}$ is referred to as the assortative case, because vertices from the same group connect with higher probability than with vertices from other groups. $c_{out} > c_{\text{in}}$ is called the disassortative case. An important conjecture [4] is that any tractable algorithm will only detect communities if

$$|c_{\text{in}} - c_{\text{out}}| > q\sqrt{c}, \tag{1}$$

where $c$ is the average degree. In the case of $q = 2$ groups, in particular, this has been rigorously proven [15, 12] (in this case, one can also prove that no algorithm could detect communities if this condition is not met). An ideal clustering algorithm should have a low computational complexity while being able to perform optimally for the stochastic block model, detecting clusters down to the transition (1).

---
[*]This work has been supported in part by the ERC under the European Union's 7th Framework Programme Grant Agreement 307087-SPARCS

So far there are two algorithms in the literature able to detect clusters down to the transition (1). One is a message-passing algorithm based on belief-propagation [5, 4]. This algorithm, however, needs to be fed with the correct parameters of the stochastic block model to perform well, and its computational complexity scales quadratically with the number of clusters, which is an important practical limitation. To avoid such problems, the most popular non-parametric approaches to clustering are spectral methods, where one classifies vertices according to the eigenvectors of a matrix associated with the network, for instance its adjacency matrix [11, 16]. However, while this works remarkably well on regular, or dense enough graphs [2], the standard versions of spectral clustering are suboptimal on graphs generated by the SBM, and in some cases completely fail to detect communities even when other (more complex) algorithms such as belief propagation can do so. Recently, a new class of spectral algorithms based on the use of a non-backtracking walk on the directed edges of the graph has been introduced [9] and argued to be better suited for spectral clustering. In particular, it has been shown to be optimal for graphs generated by the stochastic block model, and able to detect communities even in the sparse case all the way down to the theoretical limit (1).

These results are, however, not entirely satisfactory. First, the use a of a high-dimensional matrix (of dimension $2m$ - where $m$ is the number of edges - rather than $n$, the number of nodes) can be expensive, both in terms of computational time and memory. Secondly, linear algebra methods are faster and more efficient for symmetric matrices than non-symmetric ones. The first problem was partially resolved in [9] where an equivalent operator of dimensions $2n$ was shown to exist. It was still, however, a non-symmetric one and more importantly, the reduction does not extend to weighted graphs, and thus presents a strong limitation.

In this contribution, we provide the best of both worlds: a non-parametric spectral algorithm for clustering with a symmetric $n \times n$, real operator that performs as well as the non-backtracking operator of [9], in the sense that it identifies communities as soon as (1) holds. We show numerically that our approach performs as well as the belief-propagation algorithm, without needing prior knowledge of any parameter, making it the simplest algorithmically among the best-performing clustering methods. This operator is actually not new, and has been known as the Bethe Hessian in the context of statistical physics and machine learning [14, 17] or the deformed Laplacian in other fields. However, to the best of our knowledge, it has never been considered in the context of spectral clustering.

The paper is organized as follows. In Sec. 1 we give the expression of the Bethe Hessian operator. We discuss in detail its properties and its connection with both the non-backtracking operator and an Ising spin glass in Sec. 2. In Sec. 3, we study analytically the spectrum in the case of the stochastic block model. Finally, in Sec. 4 we perform numerical tests on both the stochastic block model and on some real networks.

# 1 Clustering based on the Bethe Hessian matrix

Let $\mathcal{G} = (V, E)$ be a graph with $n$ vertices, $V = \{1, ..., n\}$, and $m$ edges. Denote by $A$ its adjacency matrix, and by $D$ the diagonal matrix defined by $D_{ii} = d_i$, $\forall i \in V$, where $d_i$ is the degree of vertex $i$. We then define the Bethe Hessian matrix, sometimes called the deformed Laplacian, as

$$H(r) := (r^2 - 1)\mathbb{1} - rA + D \,, \tag{2}$$

where $|r| > 1$ is a regularizer that we will set to a well-defined value $|r| = r_c$ depending on the graph, for instance $r_c = \sqrt{c}$ in the case of the stochastic block model, where $c$ is the average degree of the graph (see Sec. 2.1).

The spectral algorithm that is the main result of this paper works as follows: we compute the eigenvectors associated with the negative eigenvalues of both $H(r_c)$ and $H(-r_c)$, and cluster them with a standard clustering algorithm such as $k$-means (or simply by looking at the sign of the components in the case of two communities). The negative eigenvalues of $H(r_c)$ reveal the assortative aspects, while those of $H(-r_c)$ reveal the disassortative ones.

Figure 1 illustrates the spectral properties of the Bethe Hessian (2) for networks generated by the stochastic block model. When $r = \pm\sqrt{c}$ the informative eigenvalues (i.e. those having eigenvectors correlated to the cluster structure) are the negative ones, while the non-informative bulk remains positive. There are as many negative eigenvalues as there are hidden clusters. It is thus straightforward to select the relevant eigenvectors. This is very unlike the situation for the operators used in standard spectral clustering algorithms (except, again, for the non-backtracking operator) where

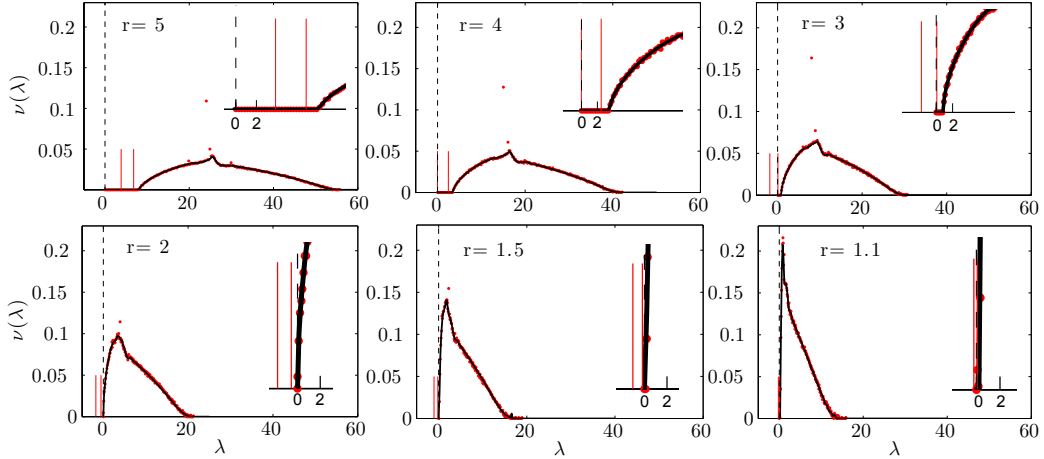

Figure 1: Spectral density of the Bethe Hessian for various values of the regularizer $r$ on the stochastic block model. The red dots are the result of the direct diagonalization of the Bethe Hessian for a graph of $10^4$ vertices with 2 clusters, with $c=4, c_{\text{in}}=7, c_{\text{out}}=1$. The black curves are the solutions to the recursion (15) for $c=4$, obtained from population dynamics (with a population of size $10^5$), see section 3. We isolated the two smallest eigenvalues, represented as small bars for convenience. The dashed black line marks the $x=0$ axis, and the inset is a zoom around this axis. At large value of $r$ (top left) $r=5$, the Bethe Hessian is positive definite and all eigenvalues are positive. As $r$ decays, the spectrum moves towards the $x=0$ axis. The smallest (non-informative) eigenvalue reaches zero for $r=c=4$ (middle top), followed, as $r$ decays further, by the second (informative) eigenvalue at $r=(c_{\text{in}}-c_{\text{out}})/2=3$, which is the value of the second largest eigenvalue of B in this case [9] (top right). Finally, the bulk reaches 0 at $r_c=\sqrt{c}=2$ (bottom left). At this point, the information is in the negative part, while the bulk is in the positive part. Interestingly, if $r$ decays further (bottom middle and right) the bulk of the spectrum remains positive, but the informative eigenvalues blend back into the bulk. The best choice is thus to work at $r_c=\sqrt{c}=2$.

one must decide in a somehow ambiguous way which eigenvalues are relevant (outside the bulk) or not (inside the bulk). Here, on the contrary, no prior knowledge of the number of communities is needed.

On more general graphs, we argue that the best choice for the regularizer is $r_c=\sqrt{\rho(\text{B})}$, where $\rho(\text{B})$ is the spectral radius of the non-backtracking operator. We support this claim both numerically, on real world networks (sec. 4.2), and analytically (sec. 3). We also show that $\rho(\text{B})$ can be computed without building the matrix B itself, by efficiently solving a quadratic eigenproblem (sec. 2.1).

The Bethe Hessian can be generalized straightforwardly to the weighed case: if the edge $(i,j)$ carries a weight $w_{ij}$, then we can use the matrix $\tilde{H}(r)$ defined by

$$\tilde{H}(r)_{ij} = \delta_{ij}\Big(1 + \sum_{k\in\partial i}\frac{w_{ik}^2}{r^2-w_{ik}^2}\Big) - \frac{rw_{ij}A_{ij}}{r^2-w_{ij}^2}\,, \tag{3}$$

where $\partial i$ denotes the set of neighbors of vertex $i$. This is in fact the general expression of the Bethe Hessian of a certain weighted statistical model (see section 2.2). If all weights are equal to unity, $\tilde{H}$ reduces to (2) up to a trivial factor. Most of the arguments developed in the following generalize immediately to $\tilde{H}$, including the relationship with the *weighted* non-backtracking operator, introduced in the conclusion of [9].

## 2  Derivation and relation to previous works

Our approach is connected to both the spectral algorithm using the non-backtracking matrix and to an Ising spin glass model. We now discuss these connections, and the properties of the Bethe Hessian operator along the way.

## 2.1 Relation with the non-backtracking matrix

The non-backtracking operator of [9] is defined as a $2m \times 2m$ non-symmetric matrix indexed by the directed edges of the graph $i \to j$

$$\mathrm{B}_{i \to j, k \to l} = \delta_{jk}(1 - \delta_{il}). \tag{4}$$

The remarkable efficiency of the non-backtracking operator is due to the particular structure of its (complex) spectrum. For graphs generated by the SBM the spectrum decomposes into a bulk of uninformative eigenvalues sharply constrained when $n \to \infty$ to the disk of radius $\sqrt{\rho(\mathrm{B})}$, where $\rho(\mathrm{B})$ is the spectral radius of B [20], well separated from the real, informative eigenvalues, that lie outside of this circle. It was also remarked that the number of real eigenvalues outside of the circle is the number of communities, when the graph was generated by the stochastic block model. More precisely, the presence of assortative communities yields real positive eigenvalues larger than $\sqrt{\rho(\mathrm{B})}$, while the presence of disassortative communities yields real negative eigenvalues smaller than $-\sqrt{\rho(\mathrm{B})}$. The authors of [9] showed that all eigenvalues $\lambda$ of B that are different from $\pm 1$ are roots of the polynomial

$$\det\left[(\lambda^2 - 1)\mathbb{1} - \lambda A + D\right] = \det H(\lambda). \tag{5}$$

This is known in graph theory as the Ihara-Bass formula for the graph zeta function. It provides the link between B and the (determinant of the) Bethe Hessian (already noticed in [23]): a real eigenvalue of B corresponds to a value of $r$ such that the Bethe Hessian has a vanishing eigenvalue.

For any finite $n$, when $r$ is large enough, $H(r)$ is positive definite. Then as $r$ decreases, a new negative eigenvalue of $H(r)$ appears when it crosses the zero axis, i.e whenever $r$ is equal to a real positive eigenvalue $\lambda$ of B. The null space of $H(\lambda)$ is related to the corresponding eigenvector of B. Denoting $(v^i)_{1 \le i \le n}$ the eigenvector of $H(\lambda)$ with eigenvalue 0, and $(v^{i \to j})_{(i,j) \in E}$ the eigenvector of B with eigenvalue $\lambda$, we have [9]:

$$v^i = \sum_{k \in \partial i} v^{k \to i}. \tag{6}$$

Therefore the vector $(v^i)_{1 \le i \le n}$ is correlated with the community structure when $(v^{i \to j})_{(i,j) \in E}$ is. The numerical experiments of section 4 show that when $r = \sqrt{c} < \lambda$, the eigenvector $(v^i)_{1 \le i \le n}$ corresponds to a strictly negative eigenvalue, and is even more correlated with the community structure than the eigenvector $(v^{i \to j})_{(i,j) \in E}$. This fact still lacks a proper theoretical understanding. We provide in section 2.2 a different, physical justification to the relevance of the "negative" eigenvectors of the Bethe Hessian for community detection. Of course, the same phenomenon takes place when increasing $r$ from a large negative value. In order to translate all the informative eigenvalues of B into negative eigenvalues of $H(r)$ we adopt

$$r_c = \sqrt{\rho(\mathrm{B})}. \tag{7}$$

since all the relevant eigenvalues of B are *outside* the circle of radius $r_c$. On the other hand, $H(r = 1)$ is the standard, positive-semidefinite, Laplacian so that for $r < r_c$, the negative eigenvalues of $H(r)$ move back into the positive part of the spectrum. This is consistent with the observation of [9] that the eigenvalues of B come in pairs having their product close to $\rho(\mathrm{B})$, so that for each root $\lambda > r_c$ of (5), corresponding to the appearance of a new negative eigenvalue, there is another root $\lambda' \simeq \rho(\mathrm{B})/\lambda < r_c$ which we numerically found to correspond to the same eigenvalue becoming positive again.

Let us stress that to compute $\rho(\mathrm{B})$, we do not need to actually build the non-backtracking matrix. First, for large random networks of a given degree distribution, $\rho(\mathrm{B}) = \langle d^2 \rangle / \langle d \rangle - 1$ [9], where $\langle d \rangle$ and $\langle d^2 \rangle$ are the first and second moments of the degree distribution. In a more general setting, we can efficiently refine this initial guess by solving for the closest root of the quadratic eigenproblem defined by (5), e.g. using a standard SLP algorithm [19]. With the choice (7), the informative eigenvalues of B are in one-to-one correspondance with the union of negative eigenvalues of $H(r_c)$ and $H(-r_c)$. Because B has as many informative eigenvalues as there are (detectable) communities in the network [9], their number will therefore tell us the number of (detectable) communities in the graph, and we will use them to infer the community membership of the nodes, by using a standard clustering algorithm such as $k$-means.

## 2.2 Hessian of the Bethe free energy

Let us define a pairwise Ising model on the graph $\mathcal{G}$ by the joint probability distribution:

$$P(\{x\}) = \frac{1}{Z} \exp\left( \sum_{(i,j)\in E} \operatorname{atanh}\left(\frac{1}{r}\right) x_i x_j \right), \tag{8}$$

where $\{x\} := \{x_i\}_{i\in\{1..n\}} \in \{\pm 1\}^n$ is a set of binary random variables sitting on the nodes of the graph $\mathcal{G}$. The regularizer $r$ is here a parameter that controls the strength of the interaction between the variables: the larger $|r|$ is, the weaker is the interaction.

In order to study this model, a standard approach in machine learning is the Bethe approximation [21] in which the means $\langle x_i \rangle$ and moments $\langle x_i x_j \rangle$ are approximated by the parameters $m_i$ and $\xi_{ij}$ that minimize the so-called Bethe free energy $F_{\text{Bethe}}(\{m_i\}, \{\xi_{ij}\})$ defined as

$$F_{\text{Bethe}}(\{m_i\}, \{\xi_{ij}\}) = - \sum_{(i,j)\in E} \operatorname{atanh}\left(\frac{1}{r}\right) \xi_{ij} + \sum_{(i,j)\in E} \sum_{x_i, x_j} \eta\left(\frac{1 + m_i x_i + m_j x_j + \xi_{ij} x_i x_j}{4}\right)$$
$$+ \sum_{i\in V} (1 - d_i) \sum_{x_i} \eta\left(\frac{1 + m_i x_i}{2}\right), \tag{9}$$

where $\eta(x) := x \ln x$. Such approach allows for instance to derive the belief propagation (BP) algorithm. Here, however, we wish to restrict to a spectral one. At very high $r$ the minimum of the Bethe free energy is given by the so-called paramagnetic point $m_i = 0$, $\xi_{ij} = \frac{1}{r}$. It turns out [14] that $m_i = 0$, $\xi_{ij} = \frac{1}{r}$ is a stationarity point of the Bethe free energy for every $r$. Instead of considering the complete Bethe free energy, we will consider only its behavior around the paramagnetic point. This can be expressed via the Hessian (matrix of second derivatives), that has been studied extensively, see e.g. [14], [17]. At the paramagnetic point, the blocks of the Hessian involving one derivative with respect to the $\xi_{ij}$ are 0, and the block involving two such derivatives is a positive definite diagonal matrix [23]. We will therefore, somewhat improperly, call Hessian the matrix

$$\mathcal{H}_{ij}(r) = \frac{\partial F_{\text{Bethe}}}{\partial m_i \partial m_j}\Big|_{m_i=0, \xi_{ij}=\frac{1}{r}}. \tag{10}$$

In particular, at the paramagnetic point:

$$\mathcal{H}(r) = \mathbb{1} + \frac{D}{r^2 - 1} - \frac{rA}{r^2 - 1} = \frac{H(r)}{r^2 - 1}. \tag{11}$$

A more general expression of the Bethe Hessian in the case of weighted interactions $\operatorname{atanh}(w_{ij}/r)$ (with weights rescaled to be in $[0,1]$) is given by eq. (3). All eigenvectors of $H(r)$ and $\mathcal{H}(r)$ are the same, as are the eigenvalues up to a multiplicative, positive factor (since we consider only $|r| > 1$).

The paramagnetic point is stable iff $H(r)$ is positive definite. The appearance of each negative eigenvalue of the Hessian corresponds to a phase transition in the Ising model at which a new cluster (or a set of clusters) starts to be identifiable. The corresponding eigenvector will give the direction towards the cluster labeling. This motivates the use of the Bethe Hessian for spectral clustering.

For tree-like graphs such as those generated by the SBM, model (8) can been studied analytically in the asymptotic limit $n \to \infty$. The location of the possible phase transitions in model (8) are also known from spin glass theory and the theory of phase transitions on random graphs (see e.g. [14, 5, 4, 17]). For positive $r$ the trivial ferromagnetic phase appears at $r = c$, while the transitions towards the phases corresponding to the hidden community structure arise between $\sqrt{c} < r < c$. For disassortative communities, the situation is symmetric with $r < -\sqrt{c}$. Interestingly, at $r = \pm\sqrt{c}$, the model undergoes a spin glass phase transition. At this point all the relevant eigenvalues have passed in the negative side (all the possible transitions from the paramagnetic states to the hidden structure have taken place) while the bulk of non-informative ones remains positive. This scenario is illustrated in Fig. 1 for the case of two assortative clusters.

## 3 The spectrum of the Bethe Hessian

The spectral density of the Bethe Hessian can be computed analytically on tree-like graphs such as those generated by the stochastic block model. This will serve two goals: i) to justify independently

our choice for the value of the regularizer $r$ and ii) to show that for all values of $r$, the bulk of uninformative eigenvalues remains in the positive region. The spectral density is defined by:

$$\nu(\lambda) = \frac{1}{n}\sum_{i=1}^{n}\delta(\lambda - \lambda_i)\,, \tag{12}$$

where the $\lambda_i$'s are the eigenvalues of the Bethe Hessian. It can be shown [18] that it is also given by

$$\nu(\lambda) = \frac{1}{\pi n}\sum_{i=1}^{n}\mathrm{Im}\Delta_i(\lambda)\,, \tag{13}$$

where the $\Delta_i$ are complex variables living on the vertices of the graph $\mathcal{G}$, which are given by:

$$\Delta_i = \left(-\lambda + r^2 + d_i - 1 - r^2\sum_{l\in\partial i}\Delta_{l\to i}\right)^{-1}\,, \tag{14}$$

where $d_i$ is the degree of node $i$ in the graph, and $\partial i$ is the set of neighbors of $i$. The $\Delta_{i\to j}$ are the (linearly stable) solution of the following belief propagation recursion, or cavity method [13],

$$\Delta_{i\to j} = \left(-\lambda + r^2 + d_i - 1 - r^2\sum_{l\in\partial i\backslash j}\Delta_{l\to i}\right)^{-1}\,. \tag{15}$$

The ingredients to derive this formula are to turn the computation of the spectral density into a marginalization problem for a graphical model on the graph $\mathcal{G}$, and then write the belief propagation equations to solve it. It can be shown [3] that this approach leads to an asymptotically exact description of the spectral density on random graphs such as those generated by the stochastic block model, which are locally tree-like in the limit where $n\to\infty$. We can solve equation (15) numerically using a population dynamics algorithm [13]: starting from a pool of variables, we iterate by drawing at each step a variable, its excess degree and its neighbors from the pool, and updating its value according to (15). The results are shown on Fig. 1: the bulk of the spectrum is always positive.

We now justify analytically that the bulk of eigenvalues of the Bethe Hessian reaches 0 at $r = \sqrt{\rho(\mathrm{B})}$. From (13) and (14), we see that if the linearly stable solution of (15) is real, then the corresponding spectral density will be equal to 0. We want to show that there exists an open set $U\subset\mathbb{R}$ around 0 in which there exists a real, stable, solution to the BP recursion. Let us call $\underline{\Delta}\in\mathbb{R}^{2m}$, where $m$ is the number of edges in $\mathcal{G}$, the vector which components are the $\Delta_{i\to j}$. We introduce the function $F:(\lambda,\underline{\Delta})\in\mathbb{R}^{2m+1}\to F(\lambda,\underline{\Delta})\in\mathbb{R}^{2m}$ defined by

$$F(\lambda,\underline{\Delta})_{i\to j} = \left(-\lambda + r^2 + d_i - 1 - r^2\sum_{l\in\partial i\backslash j}\Delta_{l\to i}\right) - \frac{1}{\Delta_{i\to j}}\,, \tag{16}$$

so that equation (15) can be rewritten as

$$F(\lambda,\underline{\Delta}) = 0\,. \tag{17}$$

It is straightforward to check that when $\lambda = 0$, the assignment $\Delta_{i\to j} = 1/r^2$ is a real solution of (17). Furthermore, the Jacobian of $F$ at this point reads

$$J_F(0,\{1/r^2\}) = \begin{pmatrix} -1 & & \\ 0 & & \\ \vdots & & r^2(r^2\mathbb{1} - B) \\ 0 & & \end{pmatrix}\,, \tag{18}$$

where B is the $2m\times 2m$ non-backtracking operator and $\mathbb{1}$ is the $2m\times 2m$ identity matrix. The square submatrix of the Jacobian containing the derivatives with respect to the messages $\Delta_{i\to j}$ is therefore invertible whenever $r > \sqrt{\rho(\mathrm{B})}$. From the continuous differentiability of $F$ around $(0,\{1/r^2\})$ and the implicit function theorem, there exists an open set $V$ containing 0 such that for all $\lambda\in V$, there exists $\tilde{\underline{\Delta}}(\lambda)\in\mathbb{R}$ solution of (17), and the function $\tilde{\underline{\Delta}}$ is continuous in $\lambda$. To show that the spectral

density is indeed 0 in an open set around $\lambda = 0$, we need to show that this solution is linearly stable. Introducing the function $G_\lambda : \underline{\Delta} \in \mathbb{R}^{2m} \to G_\lambda(\underline{\Delta}) \in \mathbb{R}^{2m}$ defined by

$$G_\lambda(\underline{\Delta})_{i \to j} = \left( -\lambda + r^2 + d_i - 1 - r^2 \sum_{l \in \partial i \backslash j} \Delta_{l \to i} \right)^{-1}, \tag{19}$$

it is enough to show that the Jacobian of $G_\lambda$ at the point $\tilde{\underline{\Delta}}(\lambda)$ has all its eigenvalues smaller than 1 in modulus, for $\lambda$ close to 0. But since $J_{G_\lambda}(\underline{\Delta})$ is continuous in $(\lambda, \underline{\Delta})$ in the neighborhood of $(0, \tilde{\underline{\Delta}}(0) = \{1/r^2\})$, and $\tilde{\underline{\Delta}}(\lambda)$ is continuous in $\lambda$, it is enough to show that the spectral radius of $J_{G_0}(\{1/r^2\})$ is smaller than 1. We compute

$$J_{G_0}(\{1/r^2\}) = \frac{1}{r^2} \mathrm{B}, \tag{20}$$

so that the spectral radius of $J_{G_0}(\{1/r^2\})$ is $\rho(\mathrm{B})/r^2$, which is (strictly) smaller than 1 as long as $r > \sqrt{\rho(\mathrm{B})}$. From the continuity of the eigenvalues of a matrix with respect to its entries, there exists an open set $U \subset V$ containing 0 such that $\forall \lambda \in U$, the solution $\tilde{\underline{\Delta}}$ of the BP recursion (15) is real, so that the corresponding spectral density in $U$ is equal to 0. This proves that the bulk of the spectrum of $H$ reaches 0 at $r = r_c = \sqrt{\rho(\mathrm{B})}$, further justifying our choice for the regularizer.

## 4 Numerical results

### 4.1 Synthetic networks

We illustrate the efficiency of the algorithm for graphs generated by the stochastic block model. Fig. 2 shows the performance of standard spectral clustering methods, as well as that of the belief propagation (BP) algorithm of [4], believed to be asymptotically optimal in large tree-like graph. The performance is measured in terms of the overlap with the true labeling, defined as

$$\left( \frac{1}{N} \sum_u \delta_{g_u, \tilde{g}_u} - \frac{1}{q} \right) \Big/ \left( 1 - \frac{1}{q} \right), \tag{21}$$

where $g_u$ is the true group label of node $u$, and $\tilde{g}_u$ is the label given by the algorithm, and we maximize over all $q!$ possible permutation of the groups. The Bethe Hessian systematically outperforms B and does almost as well as BP, which is a more complicated algorithm, that we have run here assuming the knowledge of "oracle parameters": the number of communities, their sizes, and the matrix $p_{ab}$ [5, 4]. The Bethe Hessian, on the other hand is non-parametric and infers the number of communities in the graph by counting the number of negative eigenvalues.

### 4.2 Real networks

We finally turn towards actual real graphs to illustrate the performances of our approach, and to show that even if real networks are not generated by the stochastic block model, the Bethe Hessian operator remains a useful tool. In Table 1 we give the overlap and the number of groups to be identified. We limited our experiments to this list of networks because they have known, "ground true" clusters. For each case we observed a large correlation to the ground truth, and at least equal (and sometimes better) performances with respect to the non backtracking operator. The overlap was computed assuming knowledge of the number of ground true clusters. The number of clusters is correctly given by the number of negative eigenvalues of the Bethe Hessian in all the presented cases except for the political blogs network (10 predicted clusters) and the football network (10 predicted clusters). These differences either question the statistical significance of some of the human-decided labelling, or suggest the existence of additional relevant clusters. It is also interesting to note that our approach works not only in the assortative case but also in the disassortative ones, for instance for the word adjacency networks. A Matlab implementation to reproduce the results of the Bethe Hessian for both real and synthetic networks is provided as supplementary material.

## 5 Conclusion and perspectives

We have presented here a new approach to spectral clustering using the Bethe Hessian and given evidence that this approach combines the advantages of standard sparse symmetric real matrices, with

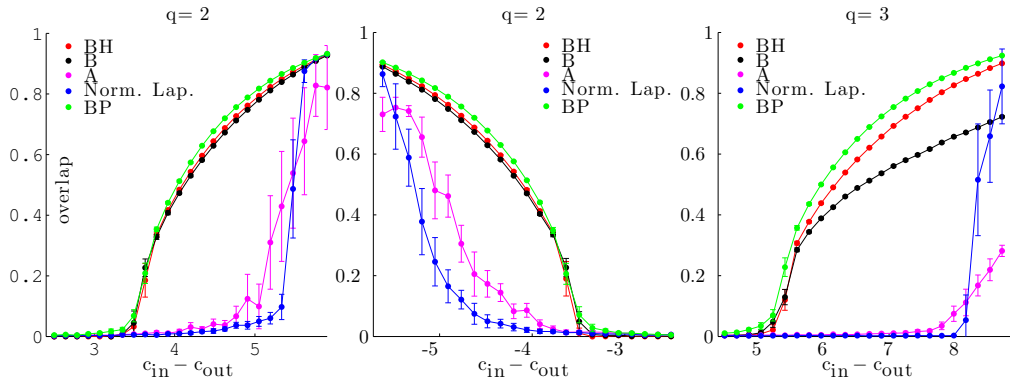

Figure 2: Performance of spectral clustering applied to graphs of size $n = 10^5$ generated from the the stochastic block model. Each point is averaged over 20 such graphs. Left: assortative case with $q = 2$ clusters (theoretical transition at 3.46); middle: disassortative case with $q = 2$ (theoretical transition at -3.46); right: assortative case with $q = 3$ clusters (theoretical transition at 5.20). For $q = 2$, we clustered according to the signs of the components of the eigenvector corresponding to the second most negative eigenvalue of the Bethe Hessian operator. For $q = 3$, we used $k$-means on the 3 "negative" eigenvectors. While both the standard adjacency (A) and symmetrically normalized Laplacian $(D^{-1/2}(D-A)D^{-1/2})$ approaches fail to identify clusters in a large relevant region, both the non-backtracking (B) and the Bethe Hessian (BH) approaches identify clusters almost as well as using the more complicated belief propagation (BP) with oracle parameters. Note, however, that the Bethe Hessian systematically outperforms the non-backtracking operator, at a smaller computational cost. Additionally, clustering with the adjacency matrix and the normalized laplacian are run on the largest connected component, while the Bethe Hessian doesn't require any kind of pre-processing of the graph. While our theory explains why clustering with the Bethe Hessian gives a positive overlap whenever clustering with B does, we currently don't have an explanation as to why the Bethe Hessian overlap is actually larger.

Table 1: Overlap for some commonly used benchmarks for community detection, computed using the signs of the second eigenvector for the networks with two communities, and using $k$-means for those with three and more communities, compared to the man-made group assignment. The non-backtracking operator detects communities in all these networks, with an overlap comparable to the performance of other spectral methods. The Bethe Hessian systematically either equals or outperforms the results obtained by the non-backtracking operator.

| PART | Non-backtracking [9] | Bethe Hessian |
|---|---|---|
| Polbooks ($q = 3$) [1] | 0.742857 | 0.757143 |
| Polblogs ($q = 2$) [10] | 0.864157 | 0.865794 |
| Karate ($q = 2$) [24] | 1 | 1 |
| Football ($q = 12$) [6] | 0.924111 | 0.924111 |
| Dolphins ($q = 2$) [16] | 0.741935 | 0.806452 |
| Adjnoun ($q = 2$) [8] | 0.625000 | 0.660714 |

the performances of the more involved non-backtracking operator, or the use of the belief propagation algorithm with oracle parameters. Advantages over other spectral methods are that the number of negative eigenvalues provides an estimate of the number of clusters, there is a well-defined way to set the parameter $r$, making the algorithm tuning-parameter free, and it is guaranteed to detect the communities generated from the stochastic block model down to the theoretical limit. This answers the quest for a tractable non-parametric approach that performs optimally in the stochastic block model. Given the large impact and the wide use of spectral clustering methods in many fields of modern science, we thus expect that our method will have a significant impact on data analysis.

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
