[Reviews · NeurIPS 2014]

Submitted by Assigned_Reviewer_27

This paper presents a spectral clustering algorithm for the sparse degree regime (degree=\theta(1)) of the stochastic blockmodel. The authors propose an alternative to the non-backtracking operator and point out that this has similar properties as the non-backtracking operator, which has been shown to be useful in the sparse regime. But the proposed data matrix is smaller than the non backtracking operator, and symmetric, therefore making eigenvalue computation easier and more accurate.

The sparse regime is indeed the hardest in terms of showing concentration of empirical eigenvalues, or performance of a clustering algorithm in the stochastic blockmodel. I think this is an interesting approach which exploits the relationship of the eigenvalues of B with the determinant of the BH matrix.

My concerns are below:

The main point of the paper was that it is better to have a symmetric matrix in terms of eigenvalue computational accuracy and efficiency. But the experiments did not make any attempt to illustrate that issue. The performance in the context of classification of the Bethe hessian is shown to be much better in Figure 2, q=3. But there is no intuition about why the bethe hessian should be better than the non-backtracking operator. While the BH matrix has negative eigenvalues corresponding to the communities, B has real eigenvalues corresponding to the communities making detection easy also.

The authors compare their method to the normalized laplacian and show its much poorer in sparse networks, but that is a well known fact because of the design of the normalized laplacian. A sparse network will have many connected components resulting in eigenvalue 1 belonging to uninformative eigenvectors, and so the normalized laplacian should be tried on the giant connected component, not on the whole graph, for fairness of comparison. If the point is that for the BH matrix no preprocessing is necessary, then that is important and should be pointed out.

The proof mainly deals with the regularization, but the fact that -the number of negative eigenvalues correspond to the number of communities- is discussed briefly through lines 179-187, while this should be expanded on. My other concern about the theory is lack of intuition in section 3, which makes it rather hard to read.

Its seems to me that Equation 3 is reduced to the unweighted matrix upto a multiplicative constant, which is only trivial because the determination of the number of communities depend on the sign of the eigenvalues which isn't affected because (r^2-1) is always positive. If this is the case, it should be made clear.

Summary: The sparse regime is indeed the hardest in terms of showing concentration of empirical eigenvalues, or performance of a clustering algorithm in the stochastic blockmodel. My concern is that the treatment seems not quite complete. In particular, questions like "why the negative eigenvalues correspond to eigenvectors with information about communities" are explained very briefly from lines 179-187. There should be more of a discussion on this.

Submitted by Assigned_Reviewer_35

This paper studies the use of the Bethe Hessian / deformed Laplacian for spectral clustering, optimised for the stochastic block model. Much of the effort in the paper is expended for showing relations with previous approaches (namely the non-backtracking matrix, and the Hessian of the Bethe free energy), and on finding good values for the parameter r in the paper.

The paper is reasonably well-written from a technical point of view. However, I found the writing to be less clear when it comes to providing a good understanding of what precisely is novel in this paper. From my understanding, this is relatively limited, as the Bethe Hessian matrix is by no means novel. I.e., as far as I understand it only the connections between its use in spectral clustering and similar approaches for clustering are novel. This is worthwhile but I do not think it will have a strong impact beyond this particular paper.

A few minor issues are:
- M has not been defined?
- Method B in figure 2 needs to be defined.
Summary: Interesting paper on the use of the Bethe Hessian matrix / deformed Laplacian for spectral clustering. However, the novelty is relatively limited.

Submitted by Assigned_Reviewer_42

Summary:

This paper proposes a new spectral approach for clustering/community detection of graphs.
It builds upon a recent approach based on the non-backtracking random walk operator whose spectral properties
allow community detection or clustering of graphs generated by the Stochastoc Block Model down to the theorectical limit.
Observing that the non-backtracking random walk operator is non-symmetric, high-dimensional and complicated, the authors of
this submission propose using a simpler, symmetric operator, the Bethe Hessian, and show that its spectrum is closely related to that of the
non-backtracking walk operator and achieves the same theoretical performance limit. The authors also derive connections
to Bethe free energy of a certain Ising model, offering perspectives from spin glass theory and phase transition of random graphs.
Experiments on synthetic networks confirm their theorectical result. On several real networks where the assumption of Stochastic Block Model
may not hold, the proposed algorithm still performs quite well.

Quality:

This work is of decent quality: the proposed approach is interesting and potentially useful in real applications, the authors
provide clear arguments about its performance guarantee and connections to other methods. The only thing that is a bit disappointing
is the experiment on real data, where details (e.g., sizes of networks, parameters of algorithms, competitors) are missing.

Clarity:

The paper is written in a succinct and clear way, but there are a few places that may benefit from
extra explanation:
* Lines 185-187: It is easy to see as r gets closer to 1, the Eigenvalues of H become non-negative because H(1) is the usual Laplacian.
But the relation between "B's Eigenvalues come in pairs having their product close to \rho(B)" and "for r < r_c, the negative eigenvalues
of H(r_c ) move back into the positive part of the spectrum" is less clear. Can the authors provide a bit more explanation?
* Line 189: What is d and what is the notation <>?
* Eq. (17): The Jacobian of F in (15) should be a (2M+1) x (2M) matrix, but Eq. (17) seems to be (2M+1) x (2M+1). It looks like
removing the first column and replacing the first row of J in (17) with -1's gives the correct Jacobian.
* Eigen vectors of Bethe Hessian: Section 2.1 gives connections of the spectrums, but how about the eigenvectors? How do the Eigenvectors
of the Bethe Hessian relate to those of the non-backtracking walk operator? How do they reveal community structures?

Originality:

The proposed approach can be viewed as an elegant reformulation of the non-backtracking walk approach, though
the main techinical result facilitating the reformulation, as pointed out by the authors, has been presented in earlier work.
Therefore it is not particularly strong in terms of originality.

Significance:

The proposed approach seems to be a promising addition to the suite of spectral clustering techniques.
Avantages over other spectral methods are that the number of negative Eigenvalues provides an estimate of the number of
clusters, there is a well-defined way to set the parameter r, making the algorithm tuning-parameter free, and it is guaranteed
to detect the communities generated by a Stochastic Block Model down to a theoretical limit.
However, whether it should be chosen over other spectral methods on real data is not clear given the experimental results in the paper.
The authors stated the performances are comparable to other spectral methods, but did not give details of the methods.
For example, how was the number of clusters chosen? Is number of negative Eigenvalues of Bethe Hessian a good estimate of number of communities?
Also, there are no timing results to demonstrate the better computational efficiency of the proposed method than the
non-backtracking approach. Most of the real networks in the paper are small, which limits the significance of the experimental results.
Summary: A decent paper with elegant and potentially useful results. Experiments on real data can be improved, but overall it is a clear accept.
Author Feedback
Author rebuttal: Dear NIPS committee,

We thank the reviewers for their valuable feedback, and their overall appreciation of our work. We will make sure to clarify the ambiguities they raised in the final version of our paper. Detailed answers to the queries of each referee are given below.

Answers to Reviewer_27:

** Computing the leading eigenvalues/eigenvectors for symmetric matrices is easier than for non-symmetric ones since the most common methods involve power iteration that often have convergence (and memory) problems for complex-valued eigenvalues. Moreover the BH matrix is smaller (by a multiplicative factor equal to the average degree) than the B matrix and its construction trivial. We think this is relatively (compared to our other claims) straightforward and hence did not include experiments illustrating that indeed the BH is computationally advantageous over B.

** In Fig. 2, in particular for q=3, the performance of BH is indeed better than for B. We did not find a theoretical reason for this. Our theory only explains that BH is able to detect when B is (i.e. the corresponding eigenvectors remain correlated). Why BH provide better overlaps remains an open problem.

** We computed the spectrum of the normalized Laplacian on the largest component (we will add this information). Hence our comparison is fair.

** We will expand on the fact that "the number of negative eigenvalues correspond to the number of communities". This follows directly from the discussion lines 179-187, but is important enough to be stressed more.

** The computation of Section 3 is technical (but standard) and we do not know of a simple way to explain intuitively.

** Unless we miss something Eq. (3) does not reduce to the unweighted case times a constant.

** "why the negative eigenvalues correspond to eigenvectors with information about communities": This is derived from the connection to the non-backtracking matrix eq. (5). For lambda being a root of (5) we have eps^i = \sum_{k \in N(i)} eps^{k\to i} to be the eigenvector of BH expressed using the eigenvector of B (i.e eps^{k\to i} is the eigenvector of B corresponding to the eigenvalue lambda, and eps^i is in the null space of H(lambda)). As we show in the numerical experiments for lambda= \sqrt(r_c) the eigenvector eps^i become even more correlated to the true groups. We will clarify this in the paper.

Answers to Reviewer_35:

This referee granted the paper with score 5. His report is brief and the principal concern is about limited novelty and potential impact of the results. We suppose this is what led to his low score.

Novelty: The Bethe Hessian is indeed a known operator (as we said),
but as far as we know it has never been suggested and tested for
clustering before, hence the very use of the BH for spectral
clustering is novel. We also think that the criticism here is slightly unfair: When the Laplacian was introduced for clustering,
nobody complained that the operator was known since the 18th century.

Impact: From the known best-performing methods (at least in the stochastic block model) the BH spectral clustering is algorithmically the simplest (fastest and not requiring parameters of the model). Having a simple and better performing method for clustering of sparse networks is a result that can be expected to have a good impact.

We will revise the introductory parts in order to clarify the above facts.

** Minor issues: B was defined in eq. (4). M is the number of edges, as defined on line 287. In fact m=M, and m was defined line 62, we will unify the notation.

Answers to Reviewer_42:

Clarity concerns:

* "B's Eigenvalues come in pairs having their product close to \rho(B)" is a fact empirically observed in [9]. For regular graphs this is be exact and easy to show. Due to eq. (5) this implies that the negative eigenvalues of BH "move back" to the positive values as r < r_c. We will explain this better.

* Those are expectations. We will clarify the notation.

* The referee is right, this is a misprint. The invertibility holds for the 2M x 2M part involving B, which is what is required in the proof.

* Eigenvectors -- see the last point in our answer to referee_27.

Originality concerns:

** The relation between the non-backtracking matrix and the Bethe Hessian was noticed in [23] in a completely different context than clustering. Our main novelty is the use of the BH for clustering.

Significance concerns:

** Experiments on real networks: We limited our experiments to the given list of networks, because for those we had information about the "ground true" clusters. We compared to spectral clustering based on the symmetrically normalized Laplacian, adjacency matrix, and modularity matrix. The number of clusters in those methods was chosen to agree with the true number of clusters. The obtained overlap was mostly worse than the one obtained by BH, but in some cases slightly better. Hence our claim about "comparable" results. The number of clusters is correctly given by the number of negative eigenvalues in all the presented cases except the political blogs network where BH predicts more clusters which possibly have significance in the given data-set, but were not included in the human-decided-labels.